# How Effectively Do People Remember Voice Disordered Speech? An Investigation of the Serial-Position Curve

**DOI:** 10.3390/brainsci8020025

**Published:** 2018-01-31

**Authors:** Scott R. Schroeder, Hannah N. Rembrandt

**Affiliations:** Department of Speech-Language-Hearing Sciences, Hofstra University, 110 Hofstra University, Hempstead, NY 11549, USA; HRembrandt1@pride.hofstra.edu

**Keywords:** memory, recall, language, voice, dysphonia, Speech-Language Pathology

## Abstract

We examined how well typical adult listeners remember the speech of a person with a voice disorder (relative to that of a person without a voice disorder). Participants (*n* = 40) listened to two lists of words (one list uttered in a disordered voice and the other list uttered in a normal voice). After each list, participants completed a free recall test, in which they tried to remember as many words as they could. While the total number of words recalled did not differ between the disordered voice condition and the normal voice condition, an investigation of the serial-position curve revealed a difference. In the normal voice condition, a parabolic (i.e., u-shaped) serial-position curve was observed, with a significant primacy effect (i.e., the beginning of the list was remembered better than the middle) and a significant recency effect (i.e., the end of the list was remembered better than the middle). In contrast, in the disordered voice condition, while there was a significant recency effect, no primacy effect was present. Thus, the increased ability to remember the first words uttered by a speaker (relative to subsequent words) may disappear when the speaker has a voice disorder. Explanations and implications of this finding are discussed.

## 1. Introduction

People differ widely in how they speak, and listeners need to adapt to these differences for spoken language processing to be efficient and accurate [1,2]. Among these speaker differences there are variations in vocal quality, with certain speakers producing excessively hoarse, rough, breathy, strained or otherwise abnormal vocal productions [3]. The need for typical listeners to adapt to voice disordered speech (i.e., dysphonic speech) is not a rare occurrence, as the lifetime prevalence of voice disorders is around 30% [4]. Despite the prevalence, very few studies have examined spoken language processing of voice disordered speech. In this study, we examined how well typical adult listeners remember words that are spoken by a speaker with a voice disorder (as compared to a normal speaker).

It can be hypothesized that words uttered in a disordered voice will be remembered with less accuracy than words uttered in a normal voice. This prediction is based on the ‘effortfulness hypothesis’ [5], which is the notion that in adverse listening situations (relative to normal listening situations), listeners will allocate more cognitive resources to identifying the spoken words, thereby leaving fewer cognitive resources left over for encoding those words into memory. On this view, a disordered voice, which one might consider an adverse listening situation, will negative affect word memory. While the ‘effortfulness hypothesis’ has not yet been applied to voice disordered speech, it can successfully account for weaker memory performance in other adverse listening situations, such as processing speech in either a noisy environment [6,7], with a hearing impairment [8], or in a second language [9].

If listening to voice disordered speech is a sufficiently adverse listening situation, then memory might be hindered, but effects might not manifest evenly across all of the words in a list. For example, the typical u-shaped serial-position curve of word recall, in which there is a primacy effect (i.e., the beginning of the list is remembered better than the middle) and a recency effect (i.e., the end of the list is remembered better than the middle), can be altered by adverse listening situations, such as background noise [10,11] and second language processing [12]. Specifically, evidence indicates that, in some cases, adverse listening situations can reduce the primacy effect and/or recency effect. Thus, it is possible that a disordered voice will negatively affect the memory advantage for the initial words and/or the final words that are heard (relative to the words in the middle of the list).

A second hypothesis, the opposite of the first, is that words spoken in a disordered voice will be remembered better than words spoken in a normal voice. This prediction is based on the ‘desirable difficulties hypothesis’ [13], which is the notion that adverse conditions (i.e., cognitively challenging situations) can lead to increased cognitive effort and consequently increased memory (though desirable difficulties might also increase memory for reasons other than effort). For example, in a study in which participants were asked to remember printed words, the words that were displayed in an inverted manner (i.e., rotated 180 degrees) were recalled better than the words that were displayed in a normal manner [14]. Thus, the more challenging task of encoding inverted words created a so-called ‘desirable difficulty’ and resulted in increased memory performance. By extension, the potentially challenging task of encoding voice disordered speech might create a desirable difficulty that boosts word memory.

In addition to these two opposing hypotheses, a third hypothesis, in which listeners remember voice disordered speech and normal speech equally well, is also possible. This third hypothesis is supported by one of the two studies that has assessed word recall for voice disordered speech versus normal speech [15]. This study found that a disordered voice did not significantly decrease word recall in children. Another study on this topic found that a disordered voice did indeed lower word recall in children [16]. However, the disordered voice and the normal voice were produced by two different speakers. The two voices therefore may have differed in a variety of dimensions beyond the vocal problem, making it difficult to attribute worse memory to the vocal problem itself.

In the current study, we tested typical adult listeners on their ability to remember lists of words that were produced in either a disordered or normal voice by the same speaker. Specifically, participants first listened to a list of words in either the speaker’s disordered voice or normal voice and then performed a free recall test for the list of words. Next, participants listened to a different set of words in the other voice condition and performed a second free recall test. In both conditions (i.e., the disordered voice condition and the normal condition), while listening to the words during encoding, participants were asked to repeat the words aloud so that word identification could be assessed. Word recall and identification were assessed for the beginning, middle, and end of the lists so that serial-position effects (i.e., primacy and recency effects) could be analyzed. By testing word recall (and identification) for a disordered voice relative to a normal voice, our understanding of the effects of voice disorders on the listener and his or her memory could be increased.

## 2. Materials and Methods

### 2.1. Participants

Forty young adults (mean age = 20.3; 31 females, 9 males) participated in the study for monetary compensation. In a questionnaire completed by participants at the end of the study, 0 participants reported a hearing impairment, and 1 participant reported a language disability (the participant noted that the language disability was related to difficulty acquiring two languages during childhood). All participants rated themselves at least a 7 (good) on a 0–10 scale in their ability to understand English (mean proficiency = 9.2; range = 7–10; standard deviation = 1.7). With at least ‘good’ proficiency in English and no current hearing or language problems, all participants were considered suitable to perform tests that required them to hear and remember English words. While all participants had ‘good’ proficiency in English (the language of the experiment), participants varied widely in their level of proficiency in a language other than English (mean proficiency in understanding a non-English language = 5.9; range = 0–10; standard deviation = 2.7). Eight of the 40 participants reported that they knew someone with a voice disorder (consistent with the high prevalence of voice disorders), and 4 of the 40 participants reported that they had taken a class in which voice disorders were discussed. In addition to the memory test for voice disordered and normal speech, participants also completed a working memory test (the digit span task from the Comprehensive Test of Phonological Processing-2). The mean percentile rank for the working memory test was 66.7%, with a standard deviation of 20.6%. Participants were primarily recruited through flyers hung up around a university campus. The flyers stated that participants were being recruited for a study on “language and cognition”. Participants gave informed consent prior to participation. The consent form and protocol were compliant with the Declaration of Helsinki and were approved by the ethics board at Hofstra University.

### 2.2. Word Memory and Identification Tests

Participants completed two versions of the memory test. In one version, participants heard a list of words spoken in a disordered voice (i.e., the ‘disordered voice condition’), whereas in the other version, participants heard a list of words spoken in a non-disordered voice (i.e., the ‘normal voice condition’). Participants were pseudo-randomized to which condition they would receive first, such that half of the participants (*n* = 20) received the disordered voice condition first, and the other half of the participants (*n* = 20) received the normal voice condition first. 

In both conditions, participants heard 15 words. Participants were instructed to repeat aloud each word immediately after hearing it. (By having participants repeat aloud each word, we could determine if the participant correctly identified the word. The experimenter marked in real-time whether the word was repeated correctly, and if not, what the participant’s response was.) Participants were also told that they would be asked to remember the words for a later memory test; thus, encoding was intentional rather than incidental. To allow sufficient time to repeat aloud each word, there was a 3 s (i.e., 3,000 millisecond) gap between words. After hearing (and repeating) all 15 words, a free recall test was given, in which participants were instructed to type in all of the words they remembered hearing. When done, participants clicked a FINISHED button on the screen. After finishing the first condition, participants went on to the second condition, which consisted of a different list of words spoken in the other voice type.

Two lists of 15 words were used (see the Appendix B for the lists of words). The two lists were randomized to the two conditions, such that list 1 was used in the disordered voice condition for half of the participants (*n* = 20) and in the normal voice condition for the other half of the participants (*n* = 20). Similarly, list 2 was used in the disordered voice condition for half of the participants (*n* = 20) and in the normal voice condition for the other half of the participants (*n* = 20). The words in the two lists did not differ from each other in number of phonemes (both lists had a mean word length of 4.7 phonemes) or in word frequency (*ps* > 0.5), as determined by the CLEARPOND database [17], though any differences between the two lists would likely impact the two conditions similarly because of the within-subjects methodology and the randomization process.

The words were presented (and the free call data (Appendix A) were collected) with SuperLab 5.0 (Cedrus Corporation, San Pedro, CA, USA) on a Macbook Pro laptop (Apple Inc., Cupertino, CA, USA). The two lists of words were peak-amplitude normalized using Audacity and were played at the same volume level across conditions and participants. Participants listened to the words through Sennheiser HD280 Pro headphones (Sennheiser, Wedemark, Germany). The words were recorded in a soundproof booth using Audacity.

The same speaker produced the words in both conditions (i.e., in the disordered voice condition and the normal voice condition). The speaker, a female college student, was a native speaker of English and did not have a voice disorder. In the normal voice condition, she uttered the words in her normal voice. In the disordered voice condition, she simulated a voice disorder. A master’s student in Speech-Language Pathology trained the speaker on how to simulate a voice disorder; specifically, the speaker was trained to produce laryngealization (i.e., vocal fry), hard glottal attacks, and strained speech, all of which are characteristic of a dysphonic voice [18,19,20,21]. To confirm that the normal voice and disordered voice were in fact normal and disordered, respectively, five graduate students who were in a Speech-Language Pathology master’s program, and who had taken a class on voice disorders, listened to the lists of words and rated the two voices on a 0–4 scale (1 = no voice disorder, 2 = mild voice disorder, 3 = moderate voice disorder, 4 = severe voice disorder). The mean scores for the normal voice and the disordered voice were 1.2 and 3.4, respectively. Thus, the normal voice received a score that was closest to no voice disorder (four of the Speech-Language Pathology students gave a 1 rating, and one gave a 2 rating), and the disordered voice received a score in the moderate-to-severe voice disorder range. 

## 3. Results

### 3.1. Word Memory

The percentage of words recalled (from 0% to 100%) at each of the three serial positions (beginning: words 1 through 5; middle: words 6 through 10; end: words 11 through 15) in the normal and disordered voice conditions were submitted to an analysis of variance (ANOVA). The ANOVA was a 2 (voice condition: normal voice vs. disordered voice) × 3 (serial position: beginning vs. middle vs. end) repeated measures ANOVA. The data that were submitted to the ANOVA are displayed in Figure 1. The ANOVA revealed no significant main effect of voice condition, *F* (1, 39) = 0.07, *p* = 0.79, a significant main effect of serial position, *F* (2, 78) = 37.88, *p* < 0.001, and no significant interaction between voice condition and serial position, *F* (2, 78) = 0.76, *p* = 0.47.

The lack of a significant main effect of voice condition (or interaction between voice condition and serial position) indicated that the percentage of words recalled did not differ between the normal voice condition and the disordered voice condition overall (or at any of the serial positions). The significant main effect of serial position was indicative of a significant primacy and/or recency effect. A follow-up Bonferroni-corrected pairwise comparison revealed a significant recency effect (*p* < 0.001), which can be clearly seen in Figure 1. In other words, collapsed across the normal voice and disordered voice conditions, the end of the list was remembered better than the middle of the list. While the recency effect was significant, a follow-up Bonferroni-corrected pairwise comparison revealed that the primacy effect was only marginally significant (*p* = 0.07). A look at the difference between the beginning of the list and the middle of the list in Figure 1 suggests why the primacy effect failed to reach full significance. That is, while there visually appears to be a primacy effect in the normal voice condition (reflecting the 12% change from beginning to middle), there does not appear to be a primacy effect in the disordered voice condition (reflecting the mere 2% change from beginning to middle). Two paired t-tests confirmed that there was a significant primacy effect (i.e., a significant difference between the beginning and middle) in the normal voice condition, *t* (39) = 2.11, *p* = 0.04, but not in the disordered voice condition, *t* (39) = 0.30, *p* = 0.77. There was a significant recency effect in both the normal voice condition, *t* (39) = 5.04, *p* < 0.001, and the disordered voice condition, *t* (39) = 5.73, *p* < 0.001.

Given that many of the participants had at least some proficiency in a language other than English (in addition to their ‘good’ English proficiency), and given that second-language experience may influence word recall, it seemed possible that the lack of a primacy effect in the disordered voice condition was due to second-language experience. To assess this possibility, we computed the Pearson’s r correlation between proficiency in a non-English language and the size of the primacy effect in the disordered voice condition. The correlation was small and not significant (*r* = −0.02, *p* = 0.90), suggesting that second-language experience was not causing the lack of a primacy effect in the disordered voice condition. Similarly, English proficiency was not significantly correlated with the primacy effect in the disordered voice condition (*r* = −0.04, *p* = 0.79).

### 3.2. Word Identification

We also examined accuracy in repeating aloud the words during the encoding phase. We referred to this as ‘word identification’. Errors in word identification included statements of non-identification (e.g., “I don’t know”), silences (i.e., participant did not say anything), substitutions (e.g., rush instead of brush), and non-words. The percentage of words correctly identified (from 0% to 100%) at each of the three serial positions in the normal and disordered voice conditions were submitted to a 2 (voice condition: normal voice vs. disordered voice) × 3 (serial position: beginning vs. middle vs. end) ANOVA. (One participant, who received the normal voice condition first and the disordered voice condition second, did not follow the instructions of repeating words aloud in the disordered voice condition, until the experimenter reminded him/her half-way through the encoding phase. This participant’s data were not included in the identification analyses.) The data that were submitted to the ANOVA are shown in Figure 2. The ANOVA revealed a significant main effect of voice condition, *F* (1, 38) = 35.42, *p* < 0.001, a marginally significant main effect of serial position, *F* (2, 76) = 2.74, *p* = 0.07, and no significant interaction between voice condition and serial position, *F* (2, 76) = 0.65, *p* = 0.53. The significant main effect of voice condition reflected lower identification accuracy in the disordered voice condition relative to the normal voice condition, as can be seen in Figure 2. The marginally significant main effect of serial position was not of particular interest and thus was not followed up (but likely reflected slightly poorer identification at the beginning of the list).

Considering the significantly lower identification in the disordered voice condition, it is possible that the lack of a primacy effect in this condition is merely an artifact of lower identification. To assess this possibility, we calculated an identification-adjusted primacy effect. In other words, the percentage recalled at each serial position was re-calculated as the number of words recalled out of the number of correctly recognized words. The same result was observed. There was still no primacy effect in the disordered voice condition, *t* (38) = 0.21, *p* = 0.84.

## 4. Discussion

This study was the first to examine how successfully typical adult listeners remember words that are spoken in a disordered voice as compared to a normal voice. We analyzed recall of words across the beginning, middle, and end of a list. For the total number of words recalled (and for number of words recalled at each serial position), words spoken in a disordered voice were statistically equal to words spoken in a normal voice. This result of no significant difference in recall between a disordered and normal voice is consistent with one of the two studies that was conducted with children [15], which also found no difference, but inconsistent with the other study that was conducted with children [16], which found worse memory for a disordered voice (although this difference could be due to the disordered and normal voices not being well-matched).

Unlike the previous studies, we analyzed serial-position effects, and interesting differences between a disordered and normal voice emerged. Specifically, we examined the u-shaped curve of recall, which is a robust and fundamental property of human memory [22,23]. The u-shaped curve reflects a primacy effect (i.e., an advantage for the beginning of the list relative to the middle) and a recency effect (i.e., an advantage for the end of the list relative to the middle). As would be expected, a u-shaped curve (with significant primacy and recency effects) was observed with the normal voice. However, the memory curve from the disordered voice was not u-shaped, as there was a significant recency effect but no significant primacy effect (i.e., no uptick on the left portion of the curve). This finding indicates that the standard memory advantage for the first words that are heard (relative to subsequent words) may not occur when the words are produced with an abnormal vocal quality. This finding cannot be attributed to lower word identification in the voice disordered condition, as the lack of a primacy effect was still observed when only accurately identified words were considered. 

Why was there no primacy effect for a disordered voice? A definite answer to this question awaits future research, but meanwhile we offer a few speculative explanations. It is possible that the lack of a primacy effect reflects a combination of the two main hypothesis laid out earlier—the ‘effortfulness hypothesis’ and the ‘desirable difficulties’ hypothesis. Recall that the ‘effortfulness hypothesis’ predicts worse memory for voice disordered speech, because the atypicality and unclarity of voice disordered speech causes the listener to expend more cognitive effort for word identification, leaving fewer cognitive resources remaining for encoding. The ‘desirable difficulties hypothesis’ predicts better memory for voice disordered speech, because the atypicality and unclarity of voice disordered speech causes the listener to expend more effort not only for word identification but also for encoding; with increased effort devoted to encoding, recall may be increased. When looking numerically at what caused the lack of a primacy effect in the disordered voice condition (see Figure 1), it appears that the beginning of the list was remembered slightly (but not significantly) worse than in the normal condition, whereas the middle of the list was remembered slightly (but not significantly) better than in the normal condition. Thus, at the beginning, the ‘effortfulness hypothesis’ may have applied. In other words, before adjusting to the unusual voice, the words may have been very difficult to identify, creating an undesirable difficulty and leaving fewer resources left over for encoding. However, after the first few words, the listener may have become better adjusted to the voice, and the ‘desirable difficulties hypothesis’ may have applied. In other words, in the middle of the list, the tasks of identification and encoding may have become easier yet were still difficult, creating a desirable difficulty that slightly boosted encoding.

For other possible explanations, we consider the traditional account of the primacy effect, which is that words at the beginning of the list are successfully transferred from short-term memory to long-term memory due to ample opportunities to rehearse the words (as opposed to words in the middle of the list, which do not get such opportunities) [22,24]. Related to this traditional account, there may have been fewer opportunities to rehearse the words in the disordered voice condition because many cognitive resources may have been devoted to word identification, leaving fewer resources for rehearsal, a possibility that is consistent with the ‘effortfulness hypothesis’. Also related to the traditional view is the possibility that the atypical phonological characteristics of the voice disordered speech may have made it more difficult to rehearse the words in the phonological loop of working memory. Still another explanation related to the traditional view is that, with their atypical phonological characteristics, the words in the disordered voice condition may have been hard to integrate into (and store in) long-term memory, as the words may have been too different from the prototypes of those words already stored in semantic long-term memory. Future research will need to determine whether these explanations or others are the cause of the absent primacy effect for voice disordered speech.

Regardless of the cause, the current findings have practical relevance. People with voice disorders (and their speech-language pathologists) may be concerned that what they say is remembered less well than what people without voice disorders say. There is a basis for this concern, as it has been found that the speech of people with other speech-language disorders, such as stuttering, is remembered less well [25]. The current findings help to alleviate this concern, as total memory was not significantly hindered by a moderate-to-severely disordered voice. Thus, teachers, lawyers, bosses, and other professionals who are likely to acquire a voice disorder through vocal over-use and whose livelihood depends on people remembering what they say need not be especially worried. Nevertheless, a voice disorder appears to alter a robust and fundamental property of human memory (i.e., the primacy effect), which may have real-world consequences for voice disordered speakers and their respective listeners. 

In conclusion, the results from the current study indicate that typical adult listeners remember the same number of words regardless of whether they are spoken in a disordered or a normal voice, but the standard memory advantage for the initial words that are heard does not seem to extend to disordered speech. These findings may be of interest to researchers, instructors, and clinicians in many different fields, including memory, speech, language, hearing, and Speech-Language Pathology. The causes and consequences of these findings will be explored in future research.

## Figures and Tables

**Figure 1 brainsci-08-00025-f001:**
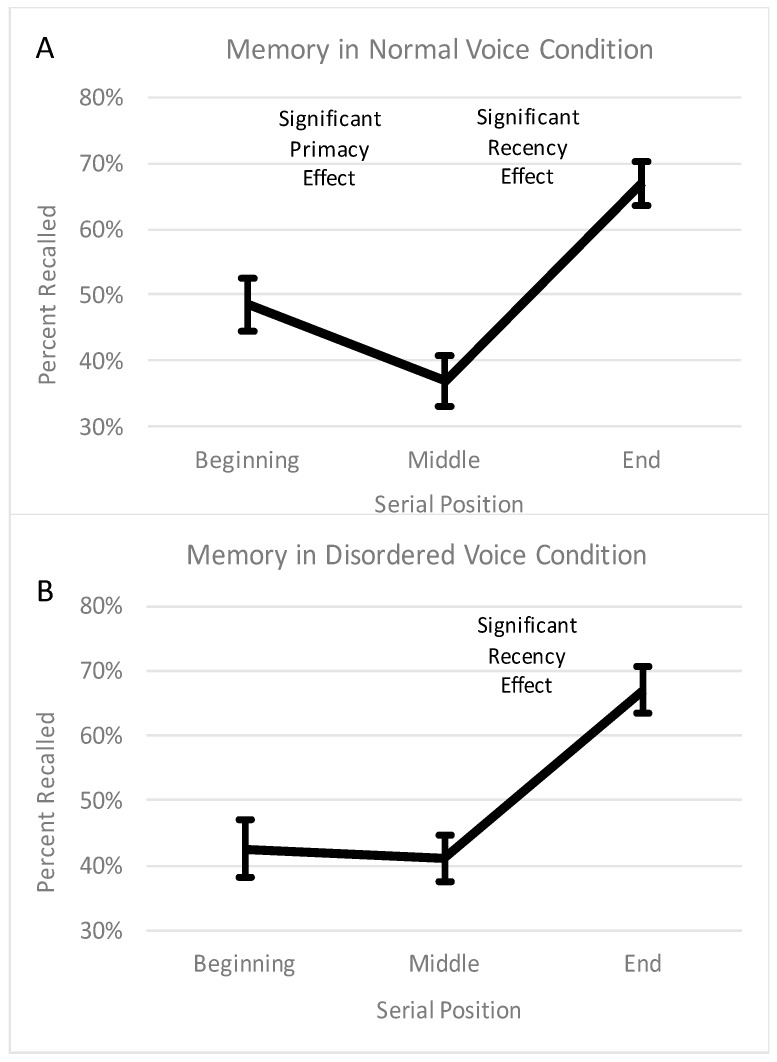
This figure depicts the percentage of words recalled in the normal voice condition (**A**) and disordered voice condition (**B**) across the beginning (first 5 words), middle (middle 5 words), and end (last 5 words) of the list. Error bars represent the standard error of the mean.

**Figure 2 brainsci-08-00025-f002:**
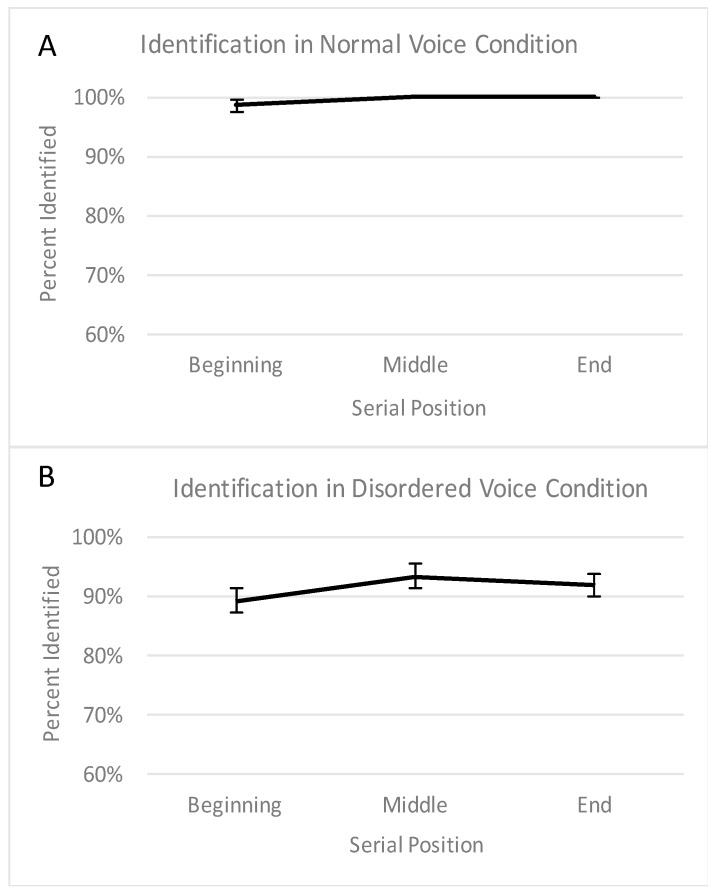
This figure depicts the percentage of words correctly identified in the normal voice condition (**A**) and disordered voice condition (**B**) across the beginning (first 5 words), middle (middle 5 words), and end (last 5 words) of the list. Error bars represent the standard error of the mean.

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
