# Peer review of "How Effectively Do People Remember Voice Disordered Speech? An Investigation of the Serial-Position Curve"

_brainsci, 2018, doi:10.3390/brainsci8020025_

Round 1

Reviewer 1 Report

The study is well-designed and the analyses of the data are appropriate. I have a couple of concerns, however. One applies to the language background of the participants: the descriptives report self-rated proficiency in English, however, it remains unclear whether these speakers are bi-lingual or not (there is actually some indication that they might be). Bilinguals process and access words differently from monolinguals, and the results might look differently, if participants are to be divided into sub-groups (e.g., bilingual vs. monolingual).

Stimuli used: The authors mention that stimuli were matched on number of letters and frequency. however, number of letters (orthography) does not matter, since the stimuli were presented auditorily. So, here we need a measure of word length on the temporal scale (how long the word lasted in ms) or laternatively number of phonemes.

Small errors in temporal forms of verbs (auxiliaries missing)

Author Response

We thank Reviewer 1 for the very helpful comments. Please see below for an explanation of how we have addressed each comment.

The reviewer asked about the participants’ bilingual (or, second-language) experience, an important variable in word processing. While all participants had high English proficiency, participants varied widely in their proficiency in a language other than English, with proficiencies ranging from 0 (none) to 10 (perfect) and with an average of 5.9 and a standard deviation of 2.7. This information is now provided in the Participants section (please see lines 93-96). We also examined whether bilingual (or, second-language) experience influenced the primary result – i.e., the lack of a primacy effect in the disordered voice condition. To this end, we computed the Pearson’s r correlation between proficiency in the second language (i.e., the non-English language) and the size of the primacy effect in the disordered voice condition. This correlation is now presented in the Results section (please see lines 188-196). (Note that this approach of treating second-language proficiency as a continuous variable was chosen over dividing participants into bilingual and monolingual sub-groups, as most participants were not clearly bilingual nor monolingual but rather somewhere in the middle.) We believe that the addition of the second-language data and analysis has improved the manuscript, and we thank Reviewer 1 for proposing the idea.

The reviewer also asked for a sound-based measure of the words, such as the number of phonemes, because the word test was auditory-based rather than visually-based. We agree that this information is important to include. The mean number of phonemes per word, as determined by the CLEARPOND database (http://clearpond.northwestern.edu), was the same for the two lists (i.e., both lists had an average word length of 4.7 phonemes). This information is now reported on lines 130-131.  

Reviewer 2 Report

The authors present their work into the classic serial positioning effects in relation to auditive word memory. Manipulation consists of the same voice actor speaking in either a normal voice or a disordered one, since earlier research findings have been ambiguous into whether voice quality will affect memory performance or not. 

The authors find that voice quality does not affect word memory in the wider sense, but that recollection of early words is harder when listening to a disordered voice. 

I found the article very easy to follow and well structured. It is also a refreshing take on what is by now a classic effect in psychology. I have very few comments (that I will outline below) and recommend the article for publication.

Line 46: I would consider getting rid of the 'the' in setence that reads:

but the effects might not manifest evenly across all of the words in a list 

Line 214: I would instead insert the 'the' in this sentence:

For the total number of words recalled 

Line 264: I would add 'have' here:

the words in the disordered voice condition may have been hard to integrate 

Footnote: The authors also state that one participant in the disordered voice condition failed the instructions and was removed. Since the experiment is counterbalanced it would be convenient for the reader to know whether this participant started with the disordered voice or with the control voice.

And thats it from me. It was a pleasure to read this manuscript.

Author Response

We thank Reviewer 2 for the valuable comments. Below, please find a description of how we have addressed each comment.

We have omitted the word ‘the’ in the sentence ‘…but the effects might not manifest evenly…’ (line 46). We have added the word ‘the’ in the sentence ‘for total number of words recalled…’ (line 227). We have also added the word ‘have’ in the sentence ‘…may been hard to integrate…’ (line 277). Lastly, we have added a note stating that the participant who failed to follow directions received the normal voice condition first and the disordered voice condition second (lines 299-300).